# Session-To-Session Variations of External Load Measures of Youth Soccer Players in Medium-Sided Games

**DOI:** 10.3390/ijerph16193612

**Published:** 2019-09-26

**Authors:** Filipe Manuel Clemente, Alireza Rabbani, Mehdi Kargarfard, Pantelis Theodoros Nikolaidis, Thomas Rosemann, Beat Knechtle

**Affiliations:** 1Polytechnic Institute of Viana do Castelo, School of Sport and Leisure, 4960-320 Melgaço, Portugal; filipe.clemente5@gmail.com; 2Institute of Tellecomunications, Covilhã Delegation, 6200-001 Covilhã, Portugal; 3Medical and Performance Department, Sporting Clube de Portugal, 2890-529 Alcochete, Portugal; alireza.rabbani@gmail.com; 4Department of Exercise Physiology, Faculty of Sports Sciences, University of Isfahan, HezarJerib str., Isfahan 81746-73441, Iran; m.kargarfard@spr.ui.ac.ir; 5Exercise Physiology Laboratory, 18450 Nikaia, Greece; pademil@hotmail.com; 6Institute of Primary Care, University of Zurich, 8091 Zurich, Switzerland; thomas.rosemann@usz.ch; 7Medbase St. Gallen Am Vadianplatz, 9001 St. Gallen, Switzerland

**Keywords:** variability, external load, training load, small-sided games

## Abstract

The aim of this study was to analyze the variability of time-motion variables during five vs. five games when completed within the same session as, and between, two different sessions. Ten under-19 male soccer players (18.27 ± 0.47 years old) participated in this study. The five vs. five matches (3 × 5 min) were played twice with a 3-day interval of rest in the same week. Moderate between-session variations were observed for TD (total distance) (range coefficient of variation (CV), 6.9; 8.3%, confidence interval (CI), (5.0; 14.0), standardized typical error (STE), 0.68; 1.06, (0.64; 1.75)) and RD (running distance) (range CV, 53.3; 145.7%, (36.6; 338.9), STE, 0.83; 1.09, (0.60; 1.76)). *PL* (player load) showed small variations (range CV, 4.9; 6.0%, [3.6; 10.1], STE, 0.37; 0.43, (0.27; 0.71)). In within-session analyses for examining the differences between sets, a small decrease was observed in RD in set 3 versus set 2 (−14.8%, 90% CI (−32.1; 6.9%); standardized difference (ES): −0.39 (0.95; 0.16)). TD decreased with moderate (−3.5%, (−6.8; −0.1%); ES: −0.65(−1.30; −0.01)) and large (−8.2%, (−11.4; −4.9%); ES: −1.58(−2.24; −0.92)) effects in sets 2 and 3, respectively, versus set 1. Our results suggest that *PL* is the most stable performance variable. It was also verified that measures had a progressive decreasing tendency within a session.

## 1. Introduction

Small-sided games (SSGs) are popular training drills that aim to reproduce the physiological, physical, and technical/tactical demands of an official soccer match [1,2,3]. Usually, SSGs are played within smaller dimensions and with fewer players than official games. Moreover, SSGs often use adjusted rules (e.g., differently sized goals, limitations on ball touches and ball possession, etc.) to modify the game for a specific purpose [4,5]. Based on the different formats (numbers of players) used by coaches, sided games can be categorized in extreme SSGs (one vs. one to two vs. two), SSGs (three vs. three to four vs. four), medium-sided games (MSGs) (five vs. five to eight vs. eight) and large-sided games (LSGs) (nine vs. nine to eleven vs. eleven) [6].

From a physiological point of view, several studies have been consistent in describing SSGs and MSGs as effective methods for improving aerobic fitness and soccer performance, considering that such games elicit intensities between 85% and 90% of maximal heart rate [6,7,8]. Such evidence is not exclusive to acute responses, considering that running-based high-intensity interval training and SSG training programs (of 6 to 12 weeks) have shown considerable improvements in aerobic capacity (by 7–8%) [9,10], lactate threshold (by 8–13%), [9,11] and high-intensity intermittent running performance (by 3–6%) [12,13]. From a physical perspective, the results are somehow different. In a recent study, it was found that formats smaller than 10 × 10 did not allow players to reach similar running intensities (total distance and high sprints per minute) compared with official matches [14]. Similar evidence was found in a study that compared SSGs, MSGs, and LSGs, and revealed that LSGs had moderately-to-largely greater values of high-intensity running and sprinting running than the other games [15]. Despite this, medium-sided games as small as four vs. four have been found to elicit greater mechanical work in players [14].

Task conditions and training regimens seem to greatly influence the proper stimuli of the physiological and physical variables [1,2,5]. However, proper stimuli are not the only concern for coaches. The variability level of the load imposed during the drills should also be considered aiming to ensure a proper stimulation of the players [16]. Regarding the acute physiological responses, in a study conducted in different SSGs and MSGs, it was found that the coefficient of variation (CV) ranged from 2.0% to 5.4% for maximal heart rate. However, it was too variable for blood lactate (10.4–43.7%) and perceived exertion (5.5–31.9%) [17]. Similar evidence was found in a study conducted using two vs. two and four vs. four games, which revealed that, in terms of heart rate responses, the reproducibility was good, but the lactate concentrations were too variable [18]. The same finding of great variability in perceived exertion was found in six vs. six games [19]. From the typical acute physiological variables, only heart rate showed good levels of reproducibility across the different formats [17,18,19,20].

Considering the reproducibility of time-motion variables, Hill-Haas et al. [16] found good reproducibility only for total distance and walking distance (0–6.9 km/h) in two vs. two, four vs. four, and six vs. six formats. Similar results were found in the six vs. six format, revealing that only total distance and metabolic power had CV values smaller than 5% [19]. In a study that compared extreme and small-sided games (one vs. one and two vs. two) with running-based drills it was possible to observe that the coefficient of variation was greater than 13% in high-speed running, very high-speed running, and sprinting distance [21]. In that study [21], less variable measures were the total distance, high and maximum accelerations, and decelerations distances. 

Also testing the reliability of under-17 players during three vs. three and four vs. three formats, good intraclass correlation values were found in total distance and number of accelerations. However, weaker results were found in distances covered between jogging and high-speed running levels and peak acceleration [22].

These findings suggest that sided games may be too variable, in terms of the physical stimulation of the players, regarding performance variables such as running distance and sprinting distance. Despite this, the studies dedicated to this topic are too small to be conclusive [18,19]. The importance of knowing whether sided games are stable in terms of producing similar stimuli within and between sessions is unquestionable. Without such information, coaches may program inefficient training regimens, thus making the response of the players to the stimulus too variable to be useful. For that reason, the purpose of this study was to examine the reproducibility of the time-motion variables during five vs. five games with small goals, when completed within the same session as, and between, two different training sessions in under-19 players. The five vs. five format was chosen based on the fact that this is typically used in soccer training and the previous studies that tested the variability did not test this format. We hypothesize that high-demanding efforts will be more variable than overall distance covered and low-intensity running.

## 2. Materials and Methods 

### 2.1. Subjects

Data were collected in ten male under-19 players (age: 18.27 ± 0.47 years old; body mass: 71.42 ± 6.89 kg; height: 177.78 ± 5.63 cm) belonging to the same team competing in a national league. Inclusion criterion for the players included participation in a minimum of 70% of the previous official matches, no injury reports in the last month, and no signs of overtraining over the two weeks prior to the study. Four defenders, four midfielders, and two forwards participated in the experiment. All participants were notified of the research procedures, requirements, benefits, and risks before signing an informed consent. The experiment followed the ethical standards of the Declaration of Helsinki. The study design was approved by a local university ethical committee with the code number ESDL.002.03.18.

### 2.2. Design

A nonexperimental descriptive comparative design was used to inspect the variations of performance variables in a medium-sided game. All data were collected during the middle-season phase (after 19 official matches and 28 weeks of training). Players had four training sessions per week and one official match during the weekend. Two 5 vs. 5 games were played in the same week with an interval of 72 h between them. The first data-collection session occurred 72 h after an official match. Performance variables were obtained using 10 Hz Global Positioning System (GPS) technology which monitored players’ movements in a valid and reliable manner [23,24]. Both games occurred at 18:15 without rainy conditions, with an ambient temperature between 14 °C and 16 °C, and with a relative humidity of 60–70%. Players were familiarized with the sided-game format during the previous week to ensure the best conditions for practice.

### 2.3. Medium-Sided Game

A pitch dimension of 30 m × 30 m (90 m^2^ per player, excluding goalkeepers) and a small goal size (2 m × 1 m) in the middle of the ending line was implemented. The regimen of the sided game was 3 × 5 min with 2 min of rest between active periods. This regimen is typically used for this kind of format [25]. Six extra soccer balls were placed around the pitch to ensure a quick continuation of the game when the ball in play left the playing area. An assistant researcher was always available to immediately replace the ball when it was kicked out. Minor rule modifications were applied (e.g., no offsides or repositioning of the ball with the foot). Games were played with coaches’ encouragement. The games were preceded by a standardized warm-up protocol including 5 min of low-intensity running, 5 min of dynamic stretching, and 5 min of short sprints and changes of direction followed by 3 min of rest. The same players and teams played the games on the same natural turf.

### 2.4. Data Collection

A 10-Hz GPS unit (including EGNOS correction, JOHAN Sports, Noordwijk, The Netherlands) and an accelerometer, gyroscope, and magnetometer (100 Hz, 3 axes, ±16 g) were used to track the players’ activity profiles. A previous study reported that this GPS sensor had a 2.5% ± 0.41% (error ± deviation) reliability for total distance covered [26]. Players wore a body-tight vest to ensure valid (e.g., body-oriented) accelerometer data. The GPS unit was then placed in a bag of the vest located in the dorsal region of the players. After the training sessions, motion data from the trackers were uploaded to the JOHAN Sports online analysis platform and then treated.

Performance variables were total distance (TD), running distance (RD = distance covered at 14–20 km/h), and player load (*PL*). *PL* was calculated as the sum of the squared rates of change in acceleration in *n* consecutive moments of the sided games on the three movement axes, where *ay* represents the acceleration in the forward-backward axis, *ax* in the sideways axis, and *az* in the vertical axis (see Equation (1)) [27]. *PL* is expressed in arbitrary units and indicates changes in players’ acceleration over time [28], which might be related to athletes’ changes of direction, impacts, and collisions throughout the games.
(1)$$PL=\sqrt{{\left(ax_{n}-ax_{n-1}\right)}^{2}+{\left(ay_{n}-ay_{n-1}\right)}^{2}+{\left(az_{n}-az_{n-1}\right)}^{2}}$$

### 2.5. Statistical Procedures

Data are presented in text, tables, and figures as either means with standard deviation (SD) or means with a 90% confidence interval (90% CI) where specified. Session-to-session variations between external load measures in each set and their accumulated values derived from all sets were analyzed by computing typical errors of measurement, expressed as coefficient of variation (CV) or as standardized units (STE), using a specifically designed spreadsheet [29]. To examine within-session variations, differences between sets and their individual differences from mean values were analyzed using standardized differences of effect size (ES) with a 90% CI [30]. The Hopkins Scale was used for interpreting ES as follows: <0.2 = trivial; 0.2–0.6 = small; 0.6–1.2 = moderate; and >1.2 = large [31]. To analyze the probability that true values were clear or trivial, a magnitude-based inference approach was used. Probabilities were computed in reference to the smallest worthwhile changes/differences (SWC, 0.2 × between-subjects SD) [32]. Qualitative probabilistic mechanistic inferences about the true effects were made using these probabilities [32]. The scale for qualitative probabilities was as follows: 25–75% = possible; 75–95% = likely; 95–99% = very likely; >99% = most likely [32]. 

## 3. Results

### 3.1. Session-to-Session Variations 

Moderate between-session variations were observed for TD (range Coefficient of variation (CV), 6.9; 8.3%, Confidence interval (CI), (5.0; 14.0), standardized typical error (STE), 0.68; 1.06, (0.64; 1.75)) and RD (range CV, 53.3; 145.7%, (36.6; 338.9), STE, 0.83; 1.09, (0.60; 1.76)) for all three sets and their accumulated values (Table 1). However, *PL* showed small variations (range CV, 4.9; 6.0%, (3.6; 10.1), STE, 0.37; 0.43, (0.27; 0.71)) in all sets and accumulated values as the most reproducible measure (Table 1). 

### 3.2. Differences Between Sets

In the first session, possibly small decreases in TD were observed in set 3 versus set 1 (−3.2%, 90%CI (−7.6; 1.4); standardized difference ES: −0.37(−0.91; 0.17)) and 2 (−2.6%, (−6.9; 1.8); ES: −0.31(−0.81; 0.2)) (Figure 1A). In the second session, a likely small decrease in TD was observed in set 2 versus set 1 (−3.7%, (−6.9; −0.5); ES: −0.46(−0.86; −0.07)), and in set 3 versus set 2 (−3.7%, (−6.9; −0.5); ES: −0.71(−1.19; −0.23)) (Figure 1A). Very likely moderate decreases in TD were also observed in set 3 versus set 1 (−9.0%, (−14.2; −3.5); ES: −1.17(−1.90; −0.44)), and in set 3 versus set 2 (−3.7%, (−6.9; −0.5); ES: −0.71(−1.19; −0.23)) in the second session (Figure 1A). When data from both sessions were averaged for each set, likely and very likely moderate decreases in TD were observed in set 2 versus set 1 (−3.5%, (−6.8; −0.1); ES: −0.65(−1.30; −0.01)), and in set 3 versus set 2 (−4.9%, (−8.1; −1.5); ES: −0.93(−1.58; −0.29)), respectively (Figure 1A). Mean TD also showed a most likely large decrease in set 3 versus set 1 (−8.2%, (−11.4; −4.9); ES: −1.58(−2.24; −0.92)) (Figure 1A).

RD showed a likely small decrease in set 3 versus set 1 (−26.0%, (−51.4; −12.7); ES: −0.42(−1.01; 0.17)) for the first session (Figure 2A). Mean RD showed a possible small decrease in set 3 versus set 2 (−14.8%, (−32.1; 6.9); ES: −0.39(−0.95; 0.16)) (Figure 2A).

The most likely moderate decreases in *PL* were observed in set 3 versus set 1 (−12.6%, (−16.7; −8.2); ES: −0.96 (−1.30; −0.61)) and versus set 2 (−10.3%, (−13.9; −6.6); ES: −0.77(−1.06; −0.49)) for the first session (Figure 3A). Possibly-to-very likely small decreases in *PL* were also found in set 2 versus set 1 (−4.5%, (−6.7; −2.3); ES: −0.28 (−0.42; 0.14)), in set 3 versus set 1 (−8.5%, (−13.0; −3.8); ES: −0.54(−0.85; −0.24)), and in set 3 versus 2 (−4.2%, (−8.8; 0.6); ES: −0.26 (−0.56; 0.04)) for the second session (Figure 3A). Mean *PL* showed possibly and very likely small decreases in set 2 versus set 1 (−3.5%, (−5.2; −1.8); ES: −0.24 (−0.36; −0.12)) and in set 3 versus set 2 (−7.3%, (−10.9; −3.6); ES: −0.52 (−0.79; −0.25)), respectively (Figure 3A). A most likely moderate decrease was also observed in mean *PL* in set 3 versus set 1 (−10.6%, (−14.3; −6.6); ES: −0.76 (−1.06; −0.47)) (Figure 3A). 

### 3.3. Differences Between Individual Sets and Mean Values Derived From of All Sets

A very likely moderate greater and lower TD was observed in set 1 (4.0%, (1.9; 6.0); ES: 0.69 (0.34; 1.04)) and set 2 (−4.5%, (−6.5; −2.5); ES: −0.82 (−1.19; −0.47)), respectively (Figure 1B). A possibly small lower RD was observed in set 3 (−10.1%, (−22.2; 3.9); ES: −0.36 (0.86; 0.13)) (Figure 2B). In *PL*, very likely small greater and lower values were observed in set 1 (4.9%, (3.2; 6.6); ES: 0.34 (0.23; 0.46)) and set 3 (−6.2%, (−8.8; −3.5); ES: −0.46 (−0.67; −0.26)), respectively (Figure 3B).

## 4. Discussion

The present study assessed the between- and within-session variability of the five vs. five sided game format for commonly used performance variables, including TD, RD, and *PL*. The main evidence revealed moderate variations in TD and RD, and small variations in *PL*, between sessions. The overall within-session analysis revealed moderate decreases in TD from the first to the second and from the second to the third set. However, large differences were found between the third and first sets. Regarding RD, small decreases were found from the second to the third set. Finally, small decreases in *PL* were found from the first to the second and from the second to the third set. However, large differences were found between the first and third sets.

Session-to-session variation was low for *PL* (CV, 4.9%; STE, 0.37) but was moderate for TD (CV, 6.9%; STE, 0.95) and RD (83.0%; STE, 1.07). In previous studies, it was found that low-intensity running (walking or jogging) and metabolic power can be reproducible across sessions but that high-intensity running and sprinting are too variable to ensure reproducibility [18,33]. In a study conducted by Rebelo et al. [33] using 6 × 6 matches, it was found that high speed and metabolic power achieved values of variability around 4%. High-speed running (>14.4 km/h) reached variability values of 13.9%. In our study, the CV of TD reached 6.9%, suggesting that coaches can be confident in developing similar distances covered for players across sessions. However, coaches cannot be confident in developing in the same level of running based on the CV value presented. Our results recommend a conservative approach to using MSGs to reproduce patterns of high intensity, considering that the great amount of variability involved can induce poor overreaching or undertraining, thus failing to achieve stabilization of the required physical demands. Possibly, smaller formats may induce a greater level of stability on the stimuli, thus being better for ensuring the proper load for players [17,18]. Our results also suggest that player load is the most reproducible variable on five vs. five between sessions, probably because player load is highly associated with total distance and not with variations in running speed [34]. Based on this, coaches must be aware that high-speed running or sprinting distances should be properly developed in specific or dedicated drills that require more stable demands in terms of the distance and frequency, or that involve adjusting conditions to help make such variables reproducible. 

In addition to observing the reproducibility of the performance variables in five vs. five matches, this study brought a new approach to training regarding variations between sets. Previous studies on SSGs mainly tested the variation of heart rate responses across different sets, revealing that the first set was significantly less intense than subsequent sets in ESSGs (extreme-sided games), SSGs, and MSGs [26,35]. However, to the best of our knowledge, only two studies have tested the variance of external load within sessions in SSGs [26,36]. The overall results of our study reveal moderate decreases in TD from the first to the second set (−3.5%) and from the first to the third set (−4.9%). Larger decreases were found from the first to the third set (−8.2%). These results can be justified by the within-exercise fatigue effect which occurred and also by the incapacity of players to manage the pacing strategies during the sets [37]. Similar to our results, in the 4 × 4 format tested by Dellal et al. [36], it was found a progressive decrease in total distance from the first to the fourth set was justified by the possible accumulation of potassium in the muscle interstitium and the subsequent depolarization of the muscle membrane potential, which reduced the force development during these intensity drills [38]. Not only can the fatigue effect explain the decreases of TD across the sets, but the training regimen and the associated recovery time could also be contributing factors. In a comparative study of two intermittent regimens (4 × 4 min and 2 × 8 min), it was found that both the first four-minute periods of exertion had greater values of total distance, thus suggesting that the time of recovery was not enough to enable players to achieve similar levels as they did in the absence of fatigue [39].

As with TD, the mean *PL* for both sessions had small decreases from the first to the second set (−3.5%) and from the second to the third (−7.3%). Also, moderate decreases from the first to the third set (−10.6%) were found. Both the fatigue effect and recovery time may have had an influence on these results. However, the strong association between TD and *PL* may justify the similarity of the results of both variables across the sets [34]. Moreover, the worst performance in terms of RD was also achieved in the third set, thus suggesting that the time of exertion, and, mainly, the length of the period of recovery and the number of sets can have a strong impact on the stabilization of high-intensity performance in this format and, probably, in the remaining SSGs, as suggested by related studies [40]. Such possibilities are not so evident in terms of heart rate response [41], thus suggesting that physical and physiological aspects must be carefully interpreted based on the lowest sensitivity of heart rate responses to the effect of accumulated fatigue in comparison to the phenomenon of temporary muscular fatigue in soccer [42].

The unique format analyzed in this study (five vs. five) and the small number of players should be considered as limitations of this study. Moreover, accelerations and decelerations must be analyzed in future MSGs because these smaller formats had a strong impact on such variables, probably mostly in sprinting or high-sprinting distance. Finally, the time of rest between the games may also contributed to a drop in efforts between sets. Despite this, this study helped us to understand how the five vs. five format can be reproducible in terms of total distance and player load, but not in terms of running distance. This should be taken into account by coaches when they are looking to establish formats to replicate the same pattern of high activity across sessions. The training regimen and recovery time should also enable a full recovery to maintain consistent within-session performances. Probably, a crossing between MSGs and specific conditions that allow for the stabilization of specific high-speed running or sprinting would optimize the training plan and the individualization of the load.

### Practical Applications

The five vs. five format can be considered reproducible for low-intensity activities and to ensure similar conditions in terms of load in different training sessionsSpecific task constraints or supplementary exercises should be used to reduce the variability of more intense running activities

## 5. Conclusions

This study concludes that five vs. five sided games can be reproducible between sessions in terms of total distance and player load, but not in terms of running distance. Regarding the variation within sessions, player load is the most stable variable considering the moderate changes in total distance and running distance between sets. An increase in the time of recovery between sets may enable a full recovery to stabilize performance during the sets. Moreover, in terms of the reproducibility of the games between sessions, some additional task conditions should be applied by coaches to ensure a low level of variability during high-intensity patterns of activity. Among others, limitation of touches on the ball, conditioning of the space, or the use of additional running-based activities should be considered to minimize the variability.

## Figures and Tables

**Figure 1 ijerph-16-03612-f001:**
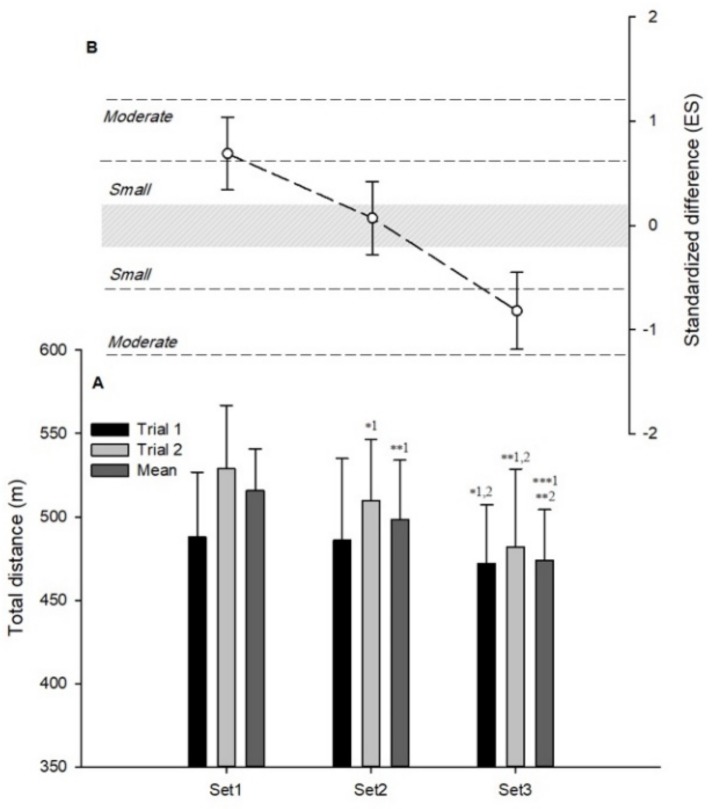
Within-session variations in total distances of small-sided games. (**A**) Mean ± standard deviation (SD) of different sets (* small, ** moderate, *** large differences; e.g., ** 1;2: Different from sets 1 and 2 with a moderate effect). (**B**) Standardized difference between individual sets and mean values derived from all sets.

**Figure 2 ijerph-16-03612-f002:**
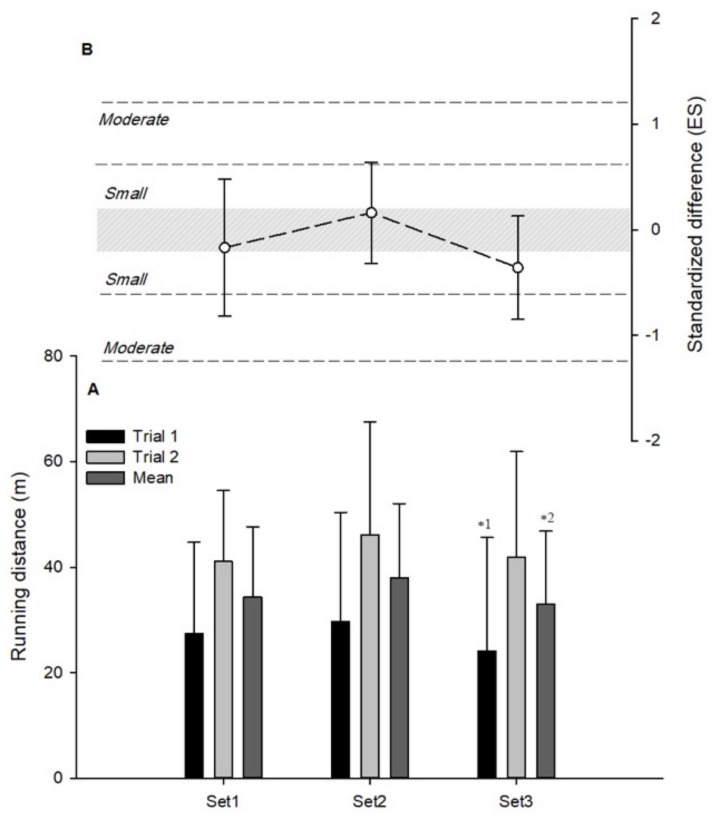
Within-session variations in running distances of small-sided games. (**A**) Mean ± SD of different sets (* small; e.g., * 1;2: Different from sets 1 and 2 with a small effect). (**B**) Standardized difference between individual sets and mean values derived from of all sets.

**Figure 3 ijerph-16-03612-f003:**
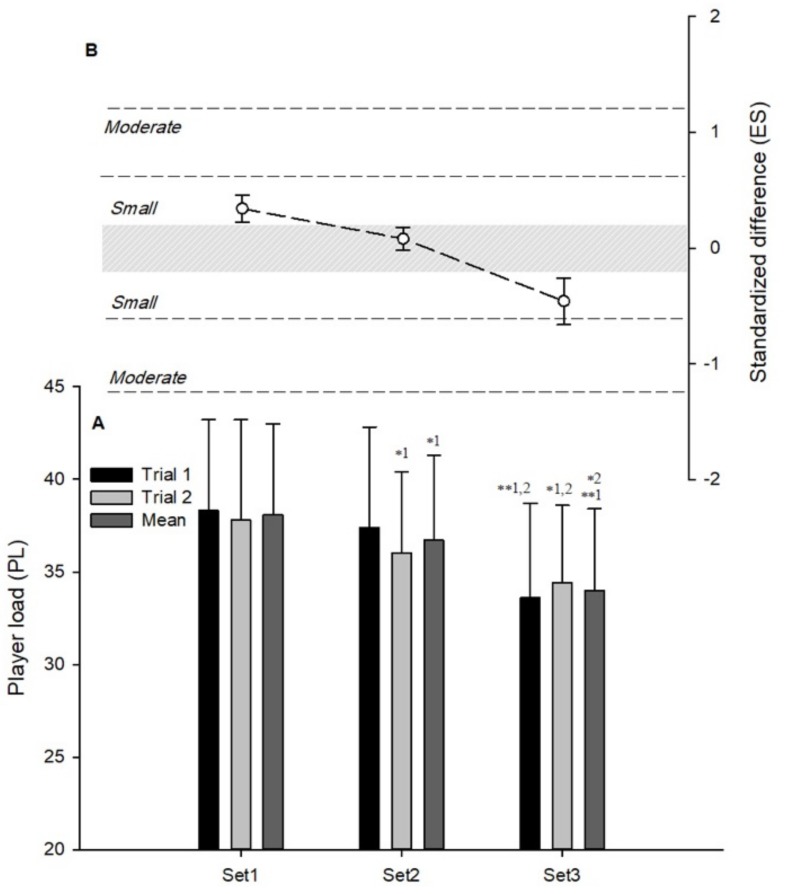
Within-session variations in player load of small-sided games. (**A**) Mean ± SD of different sets (* small, ** moderate; e.g., ** 1;2: Different from sets 1 and 2 with a moderate effect). (**B**) Standardized difference between individual sets and mean values derived from of all sets.

**Table 1 ijerph-16-03612-t001:** Between-session variations of external load measures in small-sided game (5 vs. 5 + GK).

Magnitude	Standardized Typical Error		Typical Error % (CV)		Variable	Set
	(90% CI)	Value	(90% CI)	Value		
Moderate	(0.78; 1.75)	1.06	(6.0; 14.0)	8.3	Total distance (m)	Set 1
Moderate	(0.60; 1.36)	0.83	(36.6; 102.0)	53.3	Running distance (m)	
Small	(0.30; 0.67)	0.41	(4.2; 9.8)	5.8	Player load (*PL*)	
Moderate	(0.63; 1.42)	0.68	(5.8; 13.7)	8.1	Total distance (m)	Set 2
Moderate	(0.80; 1.80)	1.09	(92.7; 338.9)	145.7	Running distance (m)	
Small	(0.32; 0.71)	0.43	(4.3; 10.1)	6	Player load (*PL*)	
Moderate	(0.64; 1.44)	0.88	(5.4; 12.7)	7.5	Total distance (m)	Set 3
Moderate	(0.75; 1.68)	1.02	(71.8; 238.9)	110	Running distance (m)	
Small	(0.31; 0.70)	0.42	(4.3; 10.1)	6	Player load (*PL*)	
Moderate	(0.70; 1.57)	0.95	(5.0; 11.6)	6.9	Total distance (m)	Sum of all sets
Moderate	(0.78; 1.76)	1.07	(55.4; 170.3)	83	Running distance (m)	
Small	(0.27; 0.60)	0.37	(3.6; 8.3)	4.9	Player load (*PL*)	

Note. CV: Coefficient of variation. CI: Confidence interval.

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
