# Peer review of "Session-To-Session Variations of External Load Measures of Youth Soccer Players in Medium-Sided Games"

_ijerph, 2019, doi:10.3390/ijerph16193612_

Round 1

Reviewer 1 Report

Dear Editor,

Firstly, thanks for allowing me to review this manuscript. We have carefully reviewed all sections of the paper (ijerph-565119). The objective of this manuscript was to analyze the variability of time-motion variables during 5 vs. 5 games when completed within the same session as, and between, two different sessions. It is interesting to study the demands of small-sided games at different conditions. The results of this study will be useful for team staff.

Respect to the manuscript, the introduction provide sufficient background and include all relevant references, the research design was appropriated, the methods were adequately described, the results were clearly presented, and the conclusions were supported by the results. For all the previous comments, my decision is accepted in the current form.

Author Response

REVIEWER 1

Dear Editor,

Firstly, thanks for allowing me to review this manuscript. We have carefully reviewed all sections of the paper (ijerph-565119). The objective of this manuscript was to analyze the variability of time-motion variables during 5 vs. 5 games when completed within the same session as, and between, two different sessions. It is interesting to study the demands of small-sided games at different conditions. The results of this study will be useful for team staff.

Respect to the manuscript, the introduction provides sufficient background and include all relevant references, the research design was appropriated, the methods were adequately described, the results were clearly presented, and the conclusions were supported by the results. For all the previous comments, my decision is accepted in the current form.

AUTHORS: DEAR REVIEWER, THANK YOU SO MUCH FOR YOUR COMMENTS.

Reviewer 2 Report

General comments

The article “Session-to-session variations of external load measures of youth soccer players in medium-sided games” aims to verify both inter and intrasession reliability of physical variables (total distance, running distance and player load) of the 5vs5 medium-sided soccer game. The subject of the article is relevant and interesting. The most relevant scientific criteria were taken into account by the authors. Besides, the results provide a very interesting discussion about the impact of the nature of the variable on the reliability, indicating that the reliability “status” may be dependent on both the sided game and the variable itself. After attending the minor changes suggested below, I am in favor of the acceptance of this article.

Specific comments

Abstract

Lines 20 and 22: please choose one (5vs5 or 5x5)

Introduction

Line 61: that “the” coefficient of variation

The introduction is clear and concise, and the most relevant information is detailed. I would recommend to include more articles that analyzed the reliability, since only two were mentioned (see below a list of interesting articles about this topic - (Ade, Harley, & Bradley, 2014; Casamichana, Castellano, & Hernández-Mendo, 2014; Dellal, Chamari, Payet, Djaoui, & Wong, 2016; Hill-Haas, Rowsell, et al., 2008; Owen, Wong, Paul, & Dellal, 2014; Stevens et al., 2016). A broader literature review would help the authors to clearly point out the novelty of the current study since this is a well-known topic in the literature.

Considering the available literature regarding the reliability of SSG, is it possible to present a hypothesis for the study?

Materials and Methods

Line 86: remove “participated in this study”.

Considering that you aimed to analyze the intersession reliability, wouldn’t it be necessary to monitor the players’ training load during the period of the data collection? If the players had a training peak during the data collection, for example, a decrease in the physical variables would be expected not because a lack of reproducibility of the SSG, but because of the fatigue effect (what, in fact, happened).

What do you think about including the ICC as a measure of reliability? This has been extensively discussed in the literature and, in my opinion, could increase the quality of the current article.

The 2-minute recovery time is enough to completely recover the players? If you want to test the intrasession reliability, players must be at the same condition in every bout. If the players start the following bout in a fatigue condition, an obvious reduction on all physical variables is expected, what does not represent a lack of reliability of the training tool.

What are the criteria for the selected variables? You should convince the reader that these present unique information besides the others available (for example, the distances in different speed zones).

Results

I would recommend the authors to enlarge the figures. In its current size, seems hard to understand all the details.

Discussion

If the authors have not controlled (or at least monitored) the training loads between the sessions, this issue must be pointed out as a limitation.

Besides this fact, the discussion is clearly the most well-written part of the article. It summarizes the most important results and provides interesting and relevant explanations for them. I have nothing else to suggest.

Suggested references

Ade, J. D., Harley, J. a., & Bradley, P. S. (2014). Physiological response, time-motion characteristics, and reproducibility of various speed-endurance drills in elite youth soccer players: Small-sided games versus generic running. International Journal of Sports Physiology and Performance, 9(3), 471–479. https://doi.org/10.1123/IJSPP.2013-0390

Bredt, S. G. T., Praça, G. M., Figueiredo, L. S., Paula, L. V., Ribeiro-Silva, P. C., Andrade, A. G. P., … Chagas, M. H. (2016). Reliability of physical, physiological and tactical measures in small-sided soccer games with numerical equality and numerical superiority. Revista Brasileira de Cineantropometria e Desempenho Humano, 18(5), 602–610. https://doi.org/10.5007/1980-0037.2016v18n5p602.

Casamichana, D., Castellano, J., & Hernández-Mendo, A. (2014). Generalizability theory applied to the study of physical profile during different small-sided games with different orientation of the field in soccer. Revista Internacional de Ciencias Del Deporte, 10(37), 194–205.

Dellal, Alexandre, Chamari, K., Payet, F., Djaoui, L., & Wong, D. P. (2016). Reproducibility of Physical Performance during Small- and Large-sided Games in Elite Soccer in Short Period: Practical Applications and Limits. Journal of Novel Physiotherapies, 6(6), 315. https://doi.org/10.4172/2165-7025.1000315

Hill-Haas, S., Coutts, A., Rowsell, G., Dawson, B., Coutts, A., & Dawson, B. (2008). Variability of acute physiological responses and performance profiles of youth soccer players in small-sided games. Journal of Science and Medicine in Sport, 11(5), 487–490. https://doi.org/10.1016/j.jsams.2007.07.006

Owen, A. L., Wong, D. P., Paul, D., & Dellal, A. (2014). Physical and technical comparisons between various-sided games within professional soccer. International Journal of Sports Medicine, 35(4), 286–292. https://doi.org/10.1055/s-0033-1351333

Stevens, T. G. A., De Ruiter, C. J., Beek, P. J., & Savelsbergh, G. J. P. (2016). Validity and reliability of 6-a-side small-sided game locomotor performance in assessing physical fitness in football players. Journal of Sports Sciences, 34(6), 527–534. https://doi.org/10.1080/02640414.2015.1116709

Author Response

REVIEWER 2

General comments

The article “Session-to-session variations of external load measures of youth soccer players in medium-sided games” aims to verify both inter and intrasession reliability of physical variables (total distance, running distance and player load) of the 5vs5 medium-sided soccer game. The subject of the article is relevant and interesting. The most relevant scientific criteria were taken into account by the authors. Besides, the results provide a very interesting discussion about the impact of the nature of the variable on the reliability, indicating that the reliability “status” may be dependent on both the sided game and the variable itself. After attending the minor changes suggested below, I am in favor of the acceptance of this article.

Specific comments

Abstract

Lines 20 and 22: please choose one (5vs5 or 5x5)

AUTHORS: DEAR REVIEWER, THANK YOU. WE HAVE CHOOSEN ONLY ONE AND MADE THE NECESSARY CHANGES ACROSS THE MANUSCRIPT.

Introduction

Line 61: that “the” coefficient of variation

AUTHORS: DEAR REVIEWER, THANK YOU. WE HAVE CHANGED ACCORDINGLY.

The introduction is clear and concise, and the most relevant information is detailed. I would recommend to include more articles that analyzed the reliability, since only two were mentioned (see below a list of interesting articles about this topic - (Ade, Harley, & Bradley, 2014; Casamichana, Castellano, & Hernández-Mendo, 2014; Dellal, Chamari, Payet, Djaoui, & Wong, 2016; Hill-Haas, Rowsell, et al., 2008; Owen, Wong, Paul, & Dellal, 2014; Stevens et al., 2016). A broader literature review would help the authors to clearly point out the novelty of the current study since this is a well-known topic in the literature.

Considering the available literature regarding the reliability of SSG, is it possible to present a hypothesis for the study?

AUTHORS: DEAR REVIEWER, THANK YOU SO MUCH. WE HAVE USED YOUR SUGGESTIONS TO IMPROVE THE RELATED WORK AND ALSO TO WRITE THE HYPOTHESIS.

Materials and Methods

Line 86: remove “participated in this study”.

AUTHORS: DEAR REVIEWER, THANK YOU. WE HAVE DELETED ACCORDINGLY.

Considering that you aimed to analyze the intersession reliability, wouldn’t it be necessary to monitor the players’ training load during the period of the data collection? If the players had a training peak during the data collection, for example, a decrease in the physical variables would be expected not because a lack of reproducibility of the SSG, but because of the fatigue effect (what, in fact, happened).

AUTHORS: DEAR REVIEWER, THANK YOU. ACTUALLY, WE HAVE COMPARED BOTH TRAINING LOAD PERIODS (BEFORE THE FIRST DATA COLLECTION AND BETWEEN THE SECOND ONE AND THE FIRST). THE COMPARISONS DID NOT REVEALED MEANINGFUL DIFFERENCES AND THAT FACT GAVE USE CONFIDENCE TO TEST THE HYPOTHESIS OF THE VARIATIONS OF THE GAMES.

What do you think about including the ICC as a measure of reliability? This has been extensively discussed in the literature and, in my opinion, could increase the quality of the current article.

AUTHORS: DEAR REVIEWER, THANK YOU. DESPITE OF POSSIBLY INTERESTING WE DO BELIEVE THAT THE TEST OF VARIABILITY WAS PROPERLY MADE BY USING THE CV.

The 2-minute recovery time is enough to completely recover the players? If you want to test the intrasession reliability, players must be at the same condition in every bout. If the players start the following bout in a fatigue condition, an obvious reduction on all physical variables is expected, what does not represent a lack of reliability of the training tool.

AUTHORS: DEAR REVIEWER, THANK YOU. ACTUALLY, OUR AIM WAS TO TEST THE VARIABILITY BETWEEN AND WITHIN SESSIONS. THE TIME PERIOD FOR RECOVERY WAS CONSIDERED BASED ON PREVIOUS RECOMMENDATIONS AND REFERENCES FOR THIS KIND OF FORMAT. HOWEVER, WE DO AGREE WITH YOUR POINT-OF-VIEW AND FOR THAT REASON WE INCLUDED THIS AS A STUDY LIMITATION IN THE DISCUSSION.

What are the criteria for the selected variables? You should convince the reader that these present unique information besides the others available (for example, the distances in different speed zones).

AUTHORS: DEAR REVIEWER, THANK YOU. WE DO AGREE WITH YOU. ACTUALLY, WE HAVE INCLUDED THE ABSENSE OF OTHER IMPORTANT VARIABLES AS STUDY LIMITATIONS IN THE DISCUSSION.

Results

I would recommend the authors to enlarge the figures. In its current size, seems hard to understand all the details.

AUTHORS: DEAR REVIEWER, THANK YOU. WE HAVE INCREASED THE SIZE.

Discussion

If the authors have not controlled (or at least monitored) the training loads between the sessions, this issue must be pointed out as a limitation.

Besides this fact, the discussion is clearly the most well-written part of the article. It summarizes the most important results and provides interesting and relevant explanations for them. I have nothing else to suggest.

AUTHORS: DEAR REVIEWER, THANK YOU. WE HAVE MONITORED THE LOADS BEFORE AND BETWEEN SESSIONS. WE WOULD LIKE TO THANK YOU FOR YOUR IMPORTANT SUGGESTIONS.

Suggested references

Ade, J. D., Harley, J. a., & Bradley, P. S. (2014). Physiological response, time-motion characteristics, and reproducibility of various speed-endurance drills in elite youth soccer players: Small-sided games versus generic running. International Journal of Sports Physiology and Performance9(3), 471–479. https://doi.org/10.1123/IJSPP.2013-0390

Bredt, S. G. T., Praça, G. M., Figueiredo, L. S., Paula, L. V., Ribeiro-Silva, P. C., Andrade, A. G. P., … Chagas, M. H. (2016). Reliability of physical, physiological and tactical measures in small-sided soccer games with numerical equality and numerical superiority. Revista Brasileira de Cineantropometria e Desempenho Humano18(5), 602–610. https://doi.org/10.5007/1980-0037.2016v18n5p602.

Casamichana, D., Castellano, J., & Hernández-Mendo, A. (2014). Generalizability theory applied to the study of physical profile during different small-sided games with different orientation of the field in soccer. Revista Internacional de Ciencias Del Deporte10(37), 194–205.

Dellal, Alexandre, Chamari, K., Payet, F., Djaoui, L., & Wong, D. P. (2016). Reproducibility of Physical Performance during Small- and Large-sided Games in Elite Soccer in Short Period: Practical Applications and Limits. Journal of Novel Physiotherapies6(6), 315. https://doi.org/10.4172/2165-7025.1000315

Hill-Haas, S., Coutts, A., Rowsell, G., Dawson, B., Coutts, A., & Dawson, B. (2008). Variability of acute physiological responses and performance profiles of youth soccer players in small-sided games. Journal of Science and Medicine in Sport11(5), 487–490. https://doi.org/10.1016/j.jsams.2007.07.006

Owen, A. L., Wong, D. P., Paul, D., & Dellal, A. (2014). Physical and technical comparisons between various-sided games within professional soccer. International Journal of Sports Medicine35(4), 286–292. https://doi.org/10.1055/s-0033-1351333

Stevens, T. G. A., De Ruiter, C. J., Beek, P. J., & Savelsbergh, G. J. P. (2016). Validity and reliability of 6-a-side small-sided game locomotor performance in assessing physical fitness in football players. Journal of Sports Sciences34(6), 527–534. https://doi.org/10.1080/02640414.2015.1116709

Reviewer 3 Report

I congratulate the authors for the general idea of the study. However, I think the problematic of the study is unclear and lacks some rationale that needs to be improved. Moreover, there are some points that it is important to clarify.

Line 51-56.

The authors consider that ".... formats smaller than 10x10 did not allow players to reach similar running intensities (total distance and high sprint per minute) compared to official 52 matches ...." My question is: is this a problem? I think that the SSG does not have to fully reflect the demands of the game, but to enhance some physical, physiological, technical and tactical aspects that the coach intends to improve.

The authors report only two external load variables. Why?

Line 58-60

"The low variability of the stimuli should also be considered to ensure that similar conditions exist across the games and conditions". Can the authors clarify the idea?

Line 60-68

It has been proven that lactate is not a reliable monitoring measure in soccer, especially in reduced games. The perception of effort is also a variable that requires careful analysis (Renfree, 2010). Thus the importance of this idea is questionable.

Line 74

Why the running distance and sprinting distance are the chosen variables

Line 79

What were the reasons that led to the choice of 5x5 format?

Line 106 and line 239. The authors should justify the choice of regime and recovery times and activity

Line 124. Why didn't the authors choose other ranges of speed?

Line 276

The authors should specify some task conditions

Line 213-223

Although the high intensity distance tends not to be reproducible in 5x5 format between sessions, what do the authors think about the game (11x11) ?? I agree that it would be important to control the training load ... But will the coach want reproducibility between sessions, or the real approach to the demands of the game?

Author Response

REVIEWER 3

I congratulate the authors for the general idea of the study. However, I think the problematic of the study is unclear and lacks some rationale that needs to be improved. Moreover, there are some points that it is important to clarify.

AUTHORS: DEAR REVIEWER, THANK YOU. WE HAVE TRIED TO IMPROVE THE MANUSCRIPT BASED ON THE REVIEWERS’ SUGGESTIONS.

Line 51-56.

The authors consider that ".... formats smaller than 10x10 did not allow players to reach similar running intensities (total distance and high sprint per minute) compared to official 52 matches ...." My question is: is this a problem? I think that the SSG does not have to fully reflect the demands of the game, but to enhance some physical, physiological, technical and tactical aspects that the coach intends to improve.

The authors report only two external load variables. Why?

AUTHORS: DEAR REVIEWER, THANK YOU. WE HAVE CHANGED THE SENTENCE TO NOT PROVIDE A BAD IDEA OF OUR RATIONALE. WE ARE NOT CRITIZING THE SSGS, ONLY TRYING TO REPORT THAT THEY ARE NOT EQUAL TO REAL MATCH. WE HAVE USED THE MEASURES TIPICALLY ASSESSED AS EXTERNAL LOAD.

Line 58-60

"The low variability of the stimuli should also be considered to ensure that similar conditions exist across the games and conditions". Can the authors clarify the idea?

AUTHORS: DEAR REVIEWER, THANK YOU. WE HAVE CHANGED THE SENTENCE.

Line 60-68

It has been proven that lactate is not a reliable monitoring measure in soccer, especially in reduced games. The perception of effort is also a variable that requires careful analysis (Renfree, 2010). Thus the importance of this idea is questionable.

AUTHORS: DEAR REVIEWER, THANK YOU. ACTUALLY, WE ARE REPORTING THE SAME. THAT THEY ARE VERY VARIABLY AND NOT RELIABLE.

Line 74

Why the running distance and sprinting distance are the chosen variables

AUTHORS: DEAR REVIEWER, THANK YOU. WE HAVE DESCRIBED THE MEASURES THAT THE LITERATURE REPORTS AS TOO VARIABLE.

Line 79

What were the reasons that led to the choice of 5x5 format?

AUTHORS: DEAR REVIEWER, THANK YOU. WE HAVE ADDED A JUSTIFICATION IN THE LAST PARAGRAPH OF THE INTRODUCTION.

Line 106 and line 239. The authors should justify the choice of regime and recovery times and activity

AUTHORS: DEAR REVIEWER, THANK YOU. WE HAVE INCLUDED A JUSTIFICATION IN THE “MEDIUM-SIDED GAME” SUB-SECTION.

Line 124. Why didn't the authors choose other ranges of speed?

AUTHORS: DEAR REVIEWER, THANK YOU. WE HAVE ADDED THIS AS A STUDY LIMITATION IN THE DISCUSSION.

Line 276

The authors should specify some task conditions

AUTHORS: DEAR REVIEWER, THANK YOU. WE HAVE ADDED SOME CONDITIONS.

Line 213-223

Although the high intensity distance tends not to be reproducible in 5x5 format between sessions, what do the authors think about the game (11x11) ?? I agree that it would be important to control the training load ... But will the coach want reproducibility between sessions, or the real approach to the demands of the game?

AUTHORS: DEAR REVIEWER, THANK YOU. ACTUALLY, THE MATCH IS VARIABLE IN NATURE. HOWEVER, THE TRAINING PROCESS SHOULD CONSIDER THE PREPARATION OF THE PLAYERS FOR BOTH PREDICTABLE AND UNPREDICTABLE DEMANDS. A PROPER STIMULATION IN TRAINING SHOULD TRY TO NORMALIZE AND ADJUST THE STIMULI AIMING TO FIT THE DRILL TO THE PURPOSE OF THE COACH. IF A HIGH VARIABILITY OCCURS, POSSIBLY, THE OBJECTIVE OF THE TASK AND THE DEVELOPMENT OF THE PLAYER WILL NOT BE ACHIEVED.
